# FedAIoT: A Federated Learning Benchmark for Artificial Intelligence of Things

## Abstract

There is a significant relevance of federated learning (FL) in the realm of Artificial Intelligence of Things (AIoT). However, most of existing FL works are not conducted on datasets collected from authentic IoT devices that capture unique modalities and inherent challenges of IoT data. In this work, we introduce `FedAIoT`, an FL benchmark for AIoT to fill this critical gap. `FedAIoT` includes eight datatsets collected from a wide range of IoT devices. These datasets cover unique IoT modalities and target representative applications of AIoT. `FedAIoT` also includes a unified end-to-end FL framework for AIoT that simplifies benchmarking the performance of the datasets. Our benchmark results shed light on the opportunities and challenges of FL for AIoT. We hope `FedAIoT` could serve as an invaluable resource to foster advancements in the important field of FL for AIoT.

## 1    Introduction

The proliferation of the Internet of Things (IoT) such as smartphones, drones, and sensors deployed at homes, as well as the gigantic amount of data they capture, have revolutionized the way we work, live, and interact with the world. The advances in Artificial Intelligence (AI) have boosted the integration of IoT and AI that turns Artificial Intelligence of Things (AIoT) into reality. However, data captured by IoT devices usually contain privacy-sensitive information. In recent years, federated learning (FL) has emerged as a privacy-preserving solution that enables extracting knowledge from the collected data while keeping the data on the devices (Kairouz et al., 2021; Wang et al., 2021).

Despite the significant relevance of FL in the realm of AIoT, most existing FL works are conducted on well-known datasets such as CIFAR-10 and CIFAR-100. *These datasets, however, do not originate from authentic IoT devices and thus fail to capture the unique modalities and inherent challenges associated with real-world IoT data.* This notable discrepancy underscores a strong need for an IoT-oriented FL benchmark to fill this critical gap.

In this work, we present `FedAIoT`, an FL benchmark for AIoT (Figure 1). At its core, `FedAIoT` includes eight well-chosen datasets collected from a wide range of IoT devices from smartwatch, smartphone and Wi-Fi routers, to drones, smart home sensors, and head-mounted device that either have already become an indispensable part of people's daily lives or are driving emerging applications. These datasets encapsulate a variety of unique IoT-specific data modalities such as wireless data, drone images, and smart home sensor data (e.g., motion, energy, humidity, temperature) that have not been explored in existing FL benchmarks. Moreover, these datasets target some of the most representative applications and innovative use cases of AIoT that are not possible with other technologies.

To facilitate the community benchmark the performance of the datasets and ensure reproducibility, `FedAIoT` includes a unified end-to-end FL framework for AIoT, which covers the complete FL-for-AIoT pipeline: from non-independent and identically distributed (non-IID) data partitioning, IoT-specific data preprocessing, to IoT-friendly models, FL hyperparameters, and IoT-factor emulator. Our framework also includes the implementations of popular schemes, models, and techniques involved in each stage of the FL-for-AIoT pipeline.

We have conducted a systematic benchmarking on the eight datasets using the end-to-end framework. Specifically, we examine the impact of varying degrees of non-IID data distributions, FL optimizers, and client sampling ratios on the performance of FL. We also evaluate the impact of noisy labels, a prevalent challenge in IoT datasets, as well as the effects of quantized training, a technique that tackles the practical limitation of resource-constrained IoT devices. Our benchmark results provide

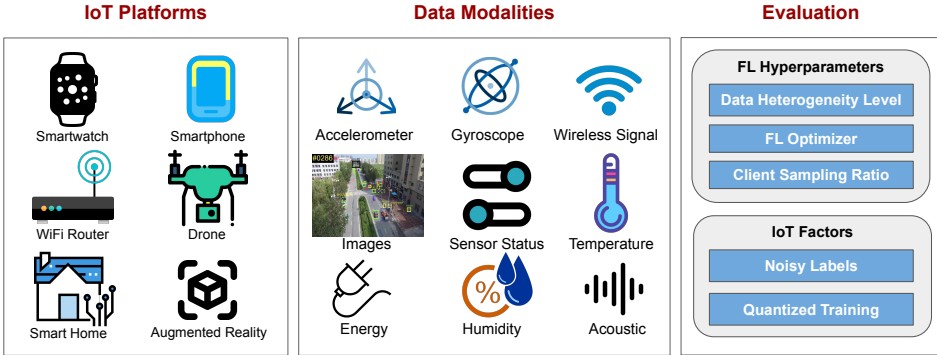

Figure 1: Overview of `FedAIoT`.

valuable insights into both the opportunities and challenges of FL for AIoT. Given the significant relevance of FL in the realm of AIoT, we hope `FedAIoT` could act as a valuable tool to foster advancements in the important area of FL for AIoT.

In summary, our work makes the following contributions.

- **IoT-focused Benchmark**. Our benchmark represents the first FL benchmark that focuses on data collected from a diverse set of authentic IoT devices. Moreover, our benchmark includes unique IoT-specific data modalities that previous benchmarks do not include.

- **Unified End-to-End FL Framework for AIoT**. We introduce the first unified end-to-end FL framework for AIoT that covers the complete FL-for-AIoT pipeline: from non-IID data partitioning, IoT-specific data preprocessing, to IoT-friendly models, FL hyperparameters, and IoT-factor emulator.

- **Novel Design of Noisy Labels**. Our benchmark also introduces a novel way to design noisy labels. Real-world FL deployments on IoT devices often encounter data labeling errors, which act as noises in the federated training. To emulate such noises, different from prior benchmarks (Feng et al., 2023; Zhang et al., 2023a) which adopt uniform error distributions, we have designed a new probabilistic label error scheme based on the insight that labels that are similar to each other are more likely to be mislabeled. This design allows us to simulate data label noises more realistically and to benchmark FL algorithms for their robustness to such noises.

- **Quantized Training**. Lastly, we are also the first FL benchmark to show the effect of quantized training on both the server and client sides in the context of FL. Previous benchmark like FLUTE (Dimitriadis et al., 2022) demonstrated the effect of quantized training during server-side aggregation for the purpose of communication reduction. In contrast, our benchmark also incorporates quantized training during client-side training to reduce the memory demands of FL on the device. This is a key difference as IoT devices are not just restrained in communication bandwidth but also in on-device memory.

## 2 RELATED WORK

The importance of data to FL research pushes the development of FL benchmarks on a variety of data modalities. Existing FL benchmarks, however, predominantly center around curating FL datasets in the domain of computer vision (CV) (Dimitriadis et al., 2022; He et al., 2021b), natural language processing (NLP) Dimitriadis et al. (2022); Lin et al. (2021), medical imaging (Terrail et al., 2022), speech and audio (Dimitriadis et al., 2022; Zhang et al., 2023a), and graph neural networks (He et al., 2021a). For example, FedCV He et al. (2021b), FedNLP (Lin et al., 2021), and FedAudio (Zhang et al., 2023a) focuses on benchmarking CV, NLP, and audio-related datasets and tasks respectively; FLUTE (Dimitriadis et al., 2022) covers a mix of datasets from CV, NLP, and audio; FLamby (Terrail et al., 2022) primarily focuses on medical imaging datasets; and FedMultimodal (Feng et al., 2023) includes multimodal datasets in the domain of emotion recognition, healthcare, multimedia, and social media. As summarized in Table 1, although these benchmarks have significantly contributed to FL research, a dedicated FL benchmark explicitly tailored for IoT data is absent. Compared to these existing FL benchmarks, `FedAIoT` is specifically designed to fill this critical gap by providing a dedicated FL benchmark on data collected from a wide range of authentic IoT devices.

Related works such as FLBench (Liang et al., 2021) and FedML (He et al., 2020) do claim AIoT support, yet neither provides benchmark results on real IoT datasets. In contrast, our `FedAIoT`

Table 1: Comparison between `FedAIoT` with existing FL benchmarks.

| | Data Type | FL Framework Designed for IoT | Noisy Labels | Quantized Training |
|---|---|---|---|---|
| FLamby | Medical Images | No | No | No |
| FedAudio | Audio Data | No | Uniform | No |
| FedMultimodal | Multimodality Data | No | Uniform | No |
| FedGraphNN | Graph Data | No | No | No |
| FedNLP | Natural Language | No | No | No |
| FLUTE | Images and Text | No | No | Server Side only |
| FedCV | Images | No | No | No |
| **FedAIoT** | **IoT Data** | **Yes** | **Probabilistic** | **Server and Client Side** |

includes 8 curated high-quality IoT datasets, offering benchmark results and analyses for these datasets. Additionally, `FedAIoT` is specifically designed for IoT, featuring modules for IoT-specific data preprocessing, IoT-friendly models, and an IoT-factor emulator.

## 3 DESIGN OF FEDAIoT

### 3.1 DATASETS

**Objective:** The objective of `FedAIoT` is to provide a benchmark that consists of high-quality and well-validated datasets collected from a wide range of IoT devices, sensor modalities, and applications. We have made significant efforts to validate a much larger pool of existing datasets and identified the high-quality ones to include in our benchmark. Table 2 provides an overview of the eight datasets included in `FedAIoT`. These datasets have diverse sizes (small: less than 7k samples; medium: 11k to 16k samples; and large: more than 60k samples). The rationale behind this design choice is to accommodate researchers who have different computing resources. For example, researchers who have very limited computing resources can still pick the relatively small datasets included in our benchmark to develop and evaluate their algorithms. In this section, we provide a brief overview of each included dataset.

**WISDM:** The Wireless Sensor Data Mining (WISDM) dataset (Lockhart et al., 2011; Weiss et al., 2019) is one of the widely used datasets for the task of daily activity recognition using accelerometer and gyroscope sensor data collected from smartphones and smartwatches. WISDM includes data collected from 51 participants performing 18 daily activities, each in a 3-minute session. We combined activities such as eating soup, chips, pasta, and sandwiches into a single category called "eating", and removed uncommon activities related to playing with balls, such as kicking, catching, or dribbling. We randomly selected 45 participants as the training set and the rest of the 6 participants were assigned to the test set. Given that the smartwatch data and smartphone data were either not collected simultaneously for most subjects or were not synchronized precisely, we partition WISDM into two independent datasets: **WISDM-W** with smartwatch data only and **WISDM-P** with smartphone data only. The total number of samples in the training and test set is 16, 569 and 4, 103 for WISDM-W and 13, 714 and 4, 073 for WISDM-P respectively. No Licence was explicitly mentioned on the dataset homepage.

**UT-HAR:** The UT-HAR dataset (Yousefi et al., 2017) is a Wi-Fi dataset for the task of contactless activity recognition. The Wi-Fi data are in the form of Channel State Information (CSI) collected using three pairs of antennas and an Intel 5300 Network Interface Card (NIC), with each antenna pair capable of capturing 30 subcarriers of CSI. UT-HAR comprises data collected from participants performing seven activities such as walking and running. UT-HAR contains a pre-determined training and test set. The total number of training and test samples is 3, 977 and 500 respectively.

**Widar:** The Widar dataset (Yang, 2020; Zheng et al., 2019) is designed for contactless gesture recognition using Wi-Fi signal strength measurements collected from strategically placed access points. This system uses an Intel 5300 NIC with $3 \times 3$ antenna pairs. The dataset features data from 17 participants showcasing 22 unique gestures, including actions like push, pull, sweeping, and clapping. To maintain consistency between training and test sets, only gestures recorded by more than three users are retained. This decision ensures the test set doesn't have data from the same participants as the training set. As a result, the balanced dataset includes nine gestures with 11, 372 samples in the training set and 5, 222 in the test set. The dataset is licensed under the Creative Commons Attribution-NonCommercial 4.0 International Licence (CC BY 4).

Table 2: Overview of the datasets included in `FedAIoT`.

| Dataset | IoT Platform | Data Modality | Data Dimension | Dataset Size | # Training Samples | # Clients |
|---------|-------------|---------------|----------------|--------------|-------------------|-----------|
| WISDM-W | Smartwatch | Accelerometer Gyroscope | $200 \times 6$ | 294 MB | $16,569$ | 80 |
| WISDM-P | Smartphone | Accelerometer Gyroscope | $200 \times 6$ | 253 MB | $13,714$ | 80 |
| UT-HAR | Wi-Fi Router | Wireless Signal | $3 \times 30 \times 250$ | 854 MB | $3,977$ | 20 |
| Widar | Wi-Fi Router | Wireless Signal | $22 \times 20 \times 20$ | 3.3 GB | $11,372$ | 40 |
| VisDrone | Drone | Images | $3 \times 224 \times 224$ | 1.8 GB | $6,471$ | 30 |
| CASAS | Smart Home | Motion Sensor Door Sensor Thermostat | $2000 \times 1$ | 233 MB | $12,190$ | 60 |
| AEP | Smart Home | Energy, Humidity Temperature | $18 \times 1$ | 12 MB | $15,788$ | 80 |
| EPIC-SOUNDS | Augmented Reality | Acoustics | $400 \times 128$ | 34 GB | $60,055$ | 210 |

**VisDrone:** The VisDrone dataset (Zhu et al., 2021) is a large-scale dataset dedicated to the task of object detection in aerial images captured by drone cameras. VisDrone includes a total of 263 video clips, which contains $179,264$ frames and $2,512,357$ labeled objects. The labeled objects fall into 12 categories (e.g., "pedestrian", "bicycle", and "car"), recorded under a variety of scenarios such as crowded urban areas, highways, and parks. The dataset contains a pre-determined training and test set. The total number of samples in the training and test set is $6,471$ and $1,610$ respectively. The dataset is licenced under Creative Commons Attribution-NonCommercial-ShareAlike 3.0 License.

**CASAS:** The CASAS dataset (Schmitter-Edgecombe & Cook, 2009), derived from the CASAS smart home project, is a smart home sensor dataset for the task of recognizing activities of daily living (ADL) based on sequences of sensor states over time to support the application of independent living. Data were collected from three distinct apartments, each equipped with three types of sensors: motion sensors, temperature sensors, and door sensors. We have selected five specific datasets from CASAS named "Milan", "Cairo", "Kyoto2", "Kyoto3", and "Kyoto4" based on the uniformity of their sensor data representation. The original ADL categories within each dataset have been consolidated into 11 categories related to home activities such as "sleep", "eat", and "bath". Activities not fitting within these categories were collectively classified as "other". The training and test set was made using an 80-20 split. Each data sample is a categorical time series of length $2,000$, representing sensor states over a certain period of time. The total number of samples in the training and test set is $12,190$ and $3,048$ respectively. No Licence was explicitly mentioned on the dataset homepage.

**AEP:** The Appliances Energy Prediction (AEP) dataset (Candanedo et al., 2017) is another smart home sensor dataset but designed for the task of home energy usage prediction. Data were collected from energy sensors, temperature sensors, and humidity sensors installed inside a home every 10 minutes over 4.5 months. The total number of samples in the training and test set is $15,788$ and $3,947$ respectively. No Licence was explicitly mentioned on the dataset homepage.

**EPIC-SOUNDS:** The EPIC-SOUNDS dataset (Huh et al., 2023) is a large-scale collection of audio recordings for the task of audio-based human activity recognition for augmented reality applications. The audio data were collected from a head-mounted microphone, containing more than $100,000$ categorized segments distributed across 44 distinct classes. The dataset contains a pre-determined training and test set. The total number of training and test samples is $60,055$ and $40,175$ respectively. The dataset is under CC BY 4 Licence.

### 3.2 END-TO-END FEDERATED LEARNING FRAMEWORK FOR AIOT

To benchmark the performance of the datasets and facilitate future research on FL for AIoT, we have designed and developed an end-to-end FL framework for AIoT as another key part of `FedAIoT`. As illustrated in Figure 2, our framework covers the complete FL-for-AIoT pipeline, including (1) non-IID data partitioning, (2) IoT-specific data preprocessing, (3) IoT-friendly models, (4) FL hyperparameters, and (5) IoT-factor emulator. In this section, we describe these components in detail.

#### 3.2.1 NON-IID DATA PARTITIONING

A key characteristic of FL is that data distribution at different clients is non-IID. The objective of non-IID data partitioning is to partition the training set such that data allocated to different clients

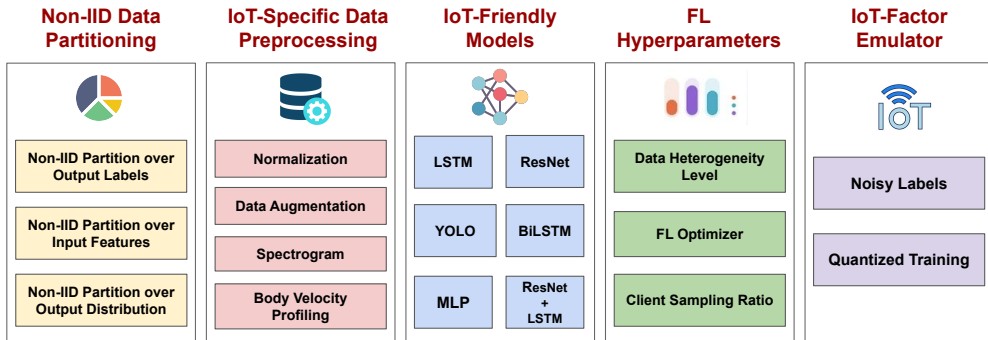

Figure 2: Overview of the end-to-end FL framework for AIoT included in `FedAIoT`.

follow the non-IID distribution. The eight datasets included in `FedAIoT` cover three fundamental tasks: classification, regression, and object detection. `FedAIoT` incorporates three different non-IID data partitioning schemes that are designed for the three tasks respectively.

**Scheme#1: Non-IID Partition over Output Labels**. For the task of classification (WISDM-W, WISDM-P, UT-HAR, Widar, CASAS, EPIC-SOUNDS) with $C$ classes, we first generate a distribution over the classes for each client by drawing from a Dirichlet distribution (See Appendix A.1.1) with parameter $\alpha$ (Hsu et al., 2019), where lower values of $\alpha$ generate more skewed distribution favoring a few classes whereas higher values of $\alpha$ result in more balanced class distributions. We use the same $\alpha$ to determine the number of samples each client receives. In addition, by drawing from a Dirichlet distribution with parameter $\alpha$, we create a distribution over the total number of samples, which is then used to allocate a varying number of samples to each client, where lower values of $\alpha$ lead to a few clients holding a majority of the samples whereas higher values of $\alpha$ create a more balanced distribution of samples across clients. Therefore, this approach allows us to generate non-IID data partitions both the class distribution and the number of samples can vary across the clients.

**Scheme#2: Non-IID Partition over Input Features**. The task of object detection (VisDrone) does not have specific classes. In such case, we use the input features to create non-IID partitions. Specifically, similar to He et al. (2021b), we first used ImageNet (Russakovsky et al., 2015) features generated from the VGG19 model (Liu & Deng, 2015), which encapsulate visual information required for subsequent analysis. With these ImageNet features as inputs, we performed clustering in the feature space using $K$-nearest neighbors to partition the dataset into clusters. Each cluster is a pseudo-class, representing a set of images sharing common visual characteristics as per the extracted ImageNet features. Lastly, Dirichlet allocation was applied on top of the pseudo-classes to create the non-IID distribution across different clients.

**Scheme#3: Non-IID Partition over Output Distribution**. For the task of regression (AEP) where output is characterized as a continuous variable, we utilize Quantile Binning (Pyle, 1999). Specifically, we divide the range of the output variable into equal groups or quantiles, ensuring that each bin accommodates roughly the same number of samples. Each category or bin is treated as a pseudo-class. Lastly, we apply Dirichlet allocation to generate the non-IID distribution of data across the clients. Note that the number of groups can be set to any value in our framework.

### 3.2.2 IoT-SPECIFIC DATA PREPROCESSING

The eight datasets included in `FedAIoT` cover diverse IoT data modalities such as wireless signals, drone images, and smart home sensor data. `FedAIoT` incorporates a suite of IoT-specific data preprocessing techniques that are designed for different IoT data modalities accordingly.

**WISDM:** We followed the standard preprocessing techniques used in accelerometer and gyroscope-based activity recognition for WISDM Ravi et al. (2005); Reyes-Ortiz et al. (2016); Ronao & Cho (2016). Specifically, for each 3-minute session, we used a 10-second sliding window with 50% overlap to extract samples from the raw accelerometer and gyroscope data sequences. We then normalize each dimension of the extracted samples by removing the mean and scaling to unit variance.

**UT-HAR:** We followed Yang et al. (2023) and applied a sliding window of 250 packets with 50% overlap to extract samples from the raw Wi-Fi data from all three antennas. We then normalize each dimension of the extracted samples by removing the mean and scaling to unit variance.

Table 3: Non-IID data partitioning schemes and models used for each dataset.

| Dataset | WISDM-W | WISDM-P | UT-HAR | Widar | VisDrone | CASAS | AEP | EPIC-SOUNDS |
|---|---|---|---|---|---|---|---|---|
| **Partition** | Output Labels | Output Labels | Output Labels | Output Labels | Input Features | Output Labels | Output Distribution | Output Labels |
| **Model** | LSTM | LSTM | ResNet18 | ResNet18 | YOLOv8n | BiLSTM | MLP | ResNet18 |

**Widar:** We first adopted the body velocity profile (BVP) processing technique as outlined in Yang et al. (2023); Zheng et al. (2019) to handle environmental variations from the data. We then applied standard scalar normalization to normalize the data. This creates data samples with the shape of $22 \times 20 \times 20$ reflecting time axis, $x$, and $y$ velocity features respectively.

**VisDrone:** We first normalized the pixel values of drone images to range from 0 to 1. Data augmentation techniques including random shifts in Hue, Saturation, and Value color space, image compression, shearing transformations, scaling transformations, horizontal and vertical flipping, and MixUp were applied to increase the diversity of the dataset.

**CASAS:** We followed Liciotti et al. (2019) to transform the sensor readings into categorical sequences, creating a form of semantic encoding. Each unique temperature setting is assigned a distinct categorical value, as are individual instances of motion sensors and door sensors being activated. For each recorded activity, we then extract a sequence of the $2,000$ previous sensor activations which is used for activity modeling and prediction.

**AEP:** Temperature data were log-transformed for skewness, and 'visibility' was binarized. Outliers below the 10th or above the 90th percentile were replaced with corresponding percentile values. Central tendency and date features were added for time-related patterns. Principal component analysis was used for data reduction, and the output was normalized using a standard scaler.

**EPIC-SOUNDS:** We first performed a Short-Time Fourier Transform (STFT) on raw audio data followed by applying a Hanning window of 10ms duration and a step size of 5ms to ensure optimal spectral resolution. We then extracted 128 Mel spectrogram features. To further enhance the data, we applied a natural logarithm scaling to the Mel spectrogram output and lastly, we padded each segment to reach a consistent length of $400$.

### 3.2.3 IOT-FRIENDLY MODELS

Since `FedAIoT` focuses on IoT devices which are resource-constrained, our choice of models is informed by a combination of model accuracies (Huh et al., 2023; Liciotti et al., 2019; Reinbothe, 2023; Seyedzadeh et al., 2018; Sholahudin et al., 2016; Terven & Cordova-Esparza, 2023; Yang et al., 2023) and model efficiency. For each included dataset, we evaluated multiple model candidates, and selected the best-performing one that adheres to the resource constraint of each IoT device listed in Table 8. As an example, for UT-HAR dataset, two model candidates (ViT and ResNet18) have similar accuracy, and we chose the more efficient ResNet18 in our benchmark. Table 3 lists the selected model for each dataset. The detail of the architecture of each model is described in Appendix A.4.

### 3.2.4 FL HYPERPARAMETERS

**Data Heterogeneity Level.** Data heterogeneity (i.e., non-IIDness) is a fundamental challenge in FL. As outlined in Section 3.2.1, `FedAIoT` facilitates the creation of diverse non-IID data partitions, which enables the simulation of different data heterogeneity levels to meet experiment requirements.

**FL Optimizer.** `FedAIoT` supports a handful of commonly used FL optimizers. In the experiment section, we showcase the benchmark results of two of the most commonly used FL optimizers: FedAvg (McMahan et al., 2017) and FedOPT Reddi et al. (2020).

**Client Sampling Ratio.** Client sampling ratio denotes the proportion of clients selected for local training in each round of FL. This hyperparameter plays a crucial role as it directly influences the computation and communication costs associated with FL. `FedAIoT` facilitates the creation of diverse client sampling ratios and the evaluation of its impact on both model performance and convergence speed during FL training.

Table 4: Overall performance.

| Dataset | Metric | Centralized | Low Data Heterogeneity ($\alpha = 0.5$) | | High Data Heterogeneity ($\alpha = 0.1$) | |
|---|---|---|---|---|---|---|
| | | | FedAvg | FedOPT | FedAvg | FedOPT |
| WISDM-W | Accuracy (%) | $74.05 \pm 2.47$ | $70.03 \pm 0.13$ | $71.50 \pm 1.52$ | $68.51 \pm 2.21$ | $65.76 \pm 2.42$ |
| WISDM-P | Accuracy (%) | $36.88 \pm 1.08$ | $36.21 \pm 0.19$ | $34.32 \pm 0.84$ | $34.28 \pm 3.28$ | $32.99 \pm 0.55$ |
| UT-HAR | Accuracy (%) | $95.24 \pm 0.75$ | $94.03 \pm 0.63$ | $94.10 \pm 0.84$ | $74.24 \pm 3.87$ | $87.78 \pm 5.48$ |
| Widar | Accuracy (%) | $61.24 \pm 0.56$ | $59.21 \pm 1.79$ | $56.26 \pm 3.11$ | $54.76 \pm 0.42$ | $47.99 \pm 3.99$ |
| VisDrone | MAP-50 (%) | $34.26 \pm 1.56$ | $32.70 \pm 1.19$ | $32.21 \pm 0.28$ | $31.23 \pm 0.70$ | $31.51 \pm 2.18$ |
| CASAS | Accuracy (%) | $83.70 \pm 2.21$ | $75.93 \pm 2.82$ | $76.40 \pm 2.20$ | $74.72 \pm 1.32$ | $75.36 \pm 2.40$ |
| AEP | $R^2$ | $0.586 \pm 0.006$ | $0.502 \pm 0.024$ | $0.503 \pm 0.011$ | $0.407 \pm 0.003$ | $0.475 \pm 0.016$ |
| EPIC-SOUNDS | Accuracy (%) | $46.97 \pm 0.24$ | $45.51 \pm 1.07$ | $42.39 \pm 2.01$ | $33.02 \pm 5.62$ | $37.21 \pm 2.68$ |

### 3.2.5 IoT-Factor Emulator

**Noisy Labels.** In real-world scenarios, IoT sensor data can be mistakenly labeled. Such data labeling errors act as noises in the federated training process. The impact of noisy labels on FL was first investigated in Zhang et al. (2023a). However, the label error distribution was considered uniform. In contrast, we propose a new probabilistic label error scheme based on the insight that labels that are similar to each other are more likely to be mislabeled. This design lets us simulate label errors in a more realistic manner. Specifically, we augment the ground truth labels of a dataset with a *confusion matrix Q* where, $Q_{ij}$ is the probability that the true label $i$ is changed to a different label $j$, i.e., $P(\hat{y} = j \mid y = i)$. The confusion matrix was constructed based on centralized training results, where the elements of $Q_{ij}$ was determined from the ratio of samples labeled as $j$ by a centrally trained model to those with ground truth label $i$.

**Quantized Training.** IoT devices are resource-constrained. Previous benchmarks like FLUTE (Dimitriadis et al., 2022) examined the performance of quantized training during server-side model aggregation for the purpose of reducing the communication cost of FL. In comparison, `FedAIoT` incorporates not only quantized model aggregation at the server side but also quantized training at the client side to reduce the memory demands of FL on the device. Quantized training on both the server and client sides is key to enabling FL for AIoT as IoT devices are not just restrained in communication bandwidth but also in on-device memory.

## 4 Experiments and Analysis

We implemented `FedAIoT` using PyTorch (Paszke et al., 2019) and Ray (Moritz et al., 2018), and conducted our experiments on NVIDIA A6000 GPUs. We run each of our experiments on three random seeds and report the mean and standard deviation.

### 4.1 Overall Performance

First, we benchmark the FL performance under two FL optimizers, FedAvg and FedOPT, under low ($\alpha = 0.5$) and high ($\alpha = 0.1$) data heterogeneity levels, and compare it against centralized training.

**Benchmark Results:** Table 4 summarizes our results. We make three observations. (1) Data heterogeneity level and FL optimizer have different impacts on different datasets. In particular, the performance of UT-HAR, AEP, and EPIC-SOUNDS is very sensitive to the data heterogeneity level. In contrast, WISDM-P, CASAS, and VisDrone show limited accuracy differences under different data heterogeneity levels. (2) Under high data heterogeneity, FedAvg has a better performance compared to FedOPT in WISDM and Widar datasets and also has lower deviation for all datasets except WISDM-P and EPIC-SOUNDS. The performance gap between the two FL optimizers reduces under low data heterogeneity. (3) Compared to the other datasets, CASAS, AEP, and WISDM-W have higher accuracy margins between centralized training and low data heterogeneity.

### 4.2 Impact of Client Sampling Ratio

Second, we benchmark the FL performance under two client sampling ratios: 10% and 30%. We report the maximum accuracy reached after completing 50%, 80%, and 100% of the total training rounds for both these ratios under high data heterogeneity, thereby offering empirical evidence of how the model performance and convergence rate are impacted by the client sampling ratio.

**Benchmark Results:** Table 5 summarizes our results. We make two observations. (1) When the client sampling ratio increases from 10% to 30%, the model performance increases at 50%, 80%, and 100% of the total training rounds across all the datasets. This demonstrates the importance of the

Table 5: Impact of client sampling ratio.

| Dataset | Training Rounds | Low Client Sampling Ratio (10%) | | | High Client Sampling Ratio (30%) | | |
|---|---|---|---|---|---|---|---|
| | | 50% Rounds | 80% Rounds | 100% Rounds | 50% Rounds | 80% Rounds | 100% Rounds |
| WISDM-W | 400 | $58.81 \pm 1.43$ | $63.82 \pm 1.53$ | $68.51 \pm 2.21$ | $65.57 \pm 2.10$ | $67.23 \pm 0.77$ | $69.21 \pm 1.13$ |
| WISDM-P | 400 | $29.49 \pm 3.65$ | $31.65 \pm 1.42$ | $34.28 \pm 3.28$ | $33.73 \pm 2.77$ | $34.01 \pm 2.27$ | $36.01 \pm 2.23$ |
| UT-HAR | 2000 | $61.81 \pm 7.01$ | $70.76 \pm 2.23$ | $74.24 \pm 3.87$ | $86.46 \pm 10.90$ | $90.84 \pm 4.42$ | $92.51 \pm 2.65$ |
| Widar | 1500 | $47.55 \pm 1.20$ | $50.65 \pm 0.24$ | $54.76 \pm 0.42$ | $53.93 \pm 2.90$ | $55.74 \pm 2.15$ | $57.39 \pm 3.14$ |
| VisDrone | 600 | $27.07 \pm 3.09$ | $31.05 \pm 1.55$ | $31.23 \pm 0.70$ | $30.56 \pm 2.71$ | $33.52 \pm 2.90$ | $34.85 \pm 0.83$ |
| CASAS | 400 | $71.68 \pm 1.96$ | $74.19 \pm 1.26$ | $74.72 \pm 1.32$ | $73.89 \pm 1.16$ | $74.68 \pm 1.50$ | $76.12 \pm 2.03$ |
| AEP | 3000 | $0.325 \pm 0.013$ | $0.371 \pm 0.017$ | $0.407 \pm 0.003$ | $0.502 \pm 0.006$ | $0.523 \pm 0.014$ | $0.538 \pm 0.005$ |
| EPIC-SOUNDS | 300 | $20.99 \pm 5.19$ | $25.73 \pm 1.99$ | $28.89 \pm 2.82$ | $23.70 \pm 6.25$ | $31.74 \pm 7.83$ | $35.11 \pm 1.99$ |

Table 6: Impact of noisy labels.

| Noisy Label Ratio | WISDM-W | WISDM-P | UT-HAR | Widar | CASAS | EPIC-SOUNDS |
|---|---|---|---|---|---|---|
| **0%** | $68.51 \pm 2.21$ | $34.28 \pm 3.28$ | $74.24 \pm 3.87$ | $54.76 \pm 0.42$ | $74.72 \pm 1.32$ | $28.89 \pm 2.82$ |
| **10%** | $50.63 \pm 4.19$ | $28.85 \pm 1.44$ | $73.75 \pm 5.67$ | $34.03 \pm 0.33$ | $65.01 \pm 2.98$ | $21.43 \pm 3.86$ |
| **30%** | $47.90 \pm 3.05$ | $27.68 \pm 0.39$ | $70.55 \pm 3.27$ | $27.20 \pm 0.56$ | $63.16 \pm 1.34$ | $13.30 \pm 0.42$ |

client sampling ratio to the model performance. (2) However, a higher sampling ratio does not always speed up model convergence to the same extent. For example, model convergence of CASAS and VisDrone are comparable at both sampling ratios whereas it is much faster for UT-HAR and AEP.

### 4.3 IMPACT OF NOISY LABELS

Next, we examine the impact of noisy labels on the FL performance under two label error ratios: 10% and 30%, and compare these results with the control scenario that involves no label errors. Note that we only showcase this for WISDM, UT-HAR, Widar, CASAS, and EPIC-SOUNDS as these are classification tasks, and the concept of noisy labels only applies to classification tasks.

**Benchmark Results:** Table 6 summarizes our results. We make two observations. (1) As the ratio of erroneous labels increases, the performance of the models decreases across all the datasets, and the impact of noisy labels varies across different datasets. For example, UT-HAR only experiences a little performance drop at 10% label error ratio, but its performance drops more when the label error ratio increases to 30%. (2) In contrast, WISDM, Widar, CASAS, and EPIC Sounds are very sensitive to label noise and show significant accuracy drop even at 10% label error ratio.

### 4.4 PERFORMANCE ON QUANTIZED TRAINING

Lastly, we examine the impact of quantized training on FL under half-precision (FP16)[1]. We assess model accuracy and memory usage under FP16 and compare the results to those from the full-precision (FP32) models. Memory usage is measured by analyzing the GPU memory usage of a model when trained with the same batch size under a centralized setting. Note that we use memory usage as the metric since it is a relatively consistent and hardware-independent metric. In contrast, other metrics such as computation speed and energy are highly hardware-dependent. Depending on the chipset that the IoT devices use, the computation speed and energy could exhibit wide variations. More importantly, new and more advanced chipsets are produced every year. The updates of the chipsets would inevitably make the benchmarking results quickly obsolete and out of date.

**Benchmark Results:** Table 7 summarizes the model performance and memory usage at two precision levels. We make three observations: (1) As expected, the memory usage significantly decreases when using FP16 precision, ranging from 57.0% to 63.3% reduction across different datasets. (2) Similar to Micikevicius et al. (2017), we also observe that model performance associated with the precision levels varies depending on the dataset. For AEP and EPIC-SOUNDS, the FP16 models improve the performance compared to the FP32 models. (3) Widar and WISDM-W have a significant decline in performance when quantized to FP16 precision. FP16 quantization also reduces the communication cost. We show the communication reduction in Table 9 in the Appendix.

---

[1]PyTorch does not support lower quantization levels like INT8 and INT4 during training as referenced Team (2023) at the time of writing and hence those were excluded.

Table 7: Performance on quantized training.

| Dataset | Metric | FP32 | | FP16 | |
|---|---|---|---|---|---|
| | | Model Performance | Memory Usage | Model Performance | Memory Usage |
| WISDM-W | Accuracy (%) | $68.51 \pm 2.21$ | 1444 MB | $60.31 \pm 5.38$ | 564 MB ($\downarrow$ 60.9%) |
| WISDM-P | Accuracy (%) | $34.28 \pm 3.28$ | 1444 MB | $30.22 \pm 2.05$ | 564 MB ($\downarrow$ 60.9%) |
| UT-HAR | Accuracy (%) | $74.24 \pm 3.87$ | 1716 MB | $72.86 \pm 4.49$ | 639 MB ($\downarrow$ 62.8%) |
| Widar | Accuracy (%) | $54.76 \pm 0.42$ | 1734 MB | $34.03 \pm 0.33$ | 636 MB ($\downarrow$ 63.3%) |
| VisDrone | MAP-50 (%) | $31.23 \pm 0.70$ | 8369 MB | $29.17 \pm 4.70$ | 3515 MB ($\downarrow$ 60.0%) |
| CASAS | Accuracy (%) | $74.72 \pm 1.32$ | 1834 MB | $72.86 \pm 4.49$ | 732 MB ($\downarrow$ 60.1%) |
| AEP | $R^2$ | $0.407 \pm 0.003$ | 1201 MB | $0.469 \pm 0.044$ | 500 MB ($\downarrow$ 58.4%) |
| EPIC-SOUNDS | Accuracy (%) | $33.02 \pm 5.62$ | 2176 MB | $35.43 \pm 6.61$ | 936 MB ($\downarrow$ 57.0%) |

## 4.5 INSIGHTS FROM BENCHMARK RESULTS

**Need for Resilience on High Data Heterogeneity:** As shown in Table 4, datasets can be sensitive to data heterogeneity. We observe that UT-HAR, Widar, AEP, and EPIC-SOUNDS show a significant impact under high data heterogeneity. These findings emphasize the need for developing advanced FL algorithms for data modalities that are sensitive to high data heterogeneity. Handling high data heterogeneity is still an open question in FL (Li et al., 2020; Zhao et al., 2018) and our benchmark shows that it is a limiting factor for some of the IoT data modalities as well. To mitigate high data heterogeneity, although many techniques have been proposed (Arivazhagan et al., 2019; Sattler et al., 2019), their performance is still constrained. Most recently, some exploration on incorporating generative pre-trained transformers (GPT) as part of the FL framework has shown great performance in mitigating high data heterogeneity (Zhang et al., 2023b).

**Need for Balancing between Client Sampling Ratio and Resource Consumption of IoT Devices:** Table 5 reveals that a higher sampling ratio can lead to improved performance in the long run. However, higher client sampling ratios generally entail increased communication overheads and energy consumption, which may not be desirable for IoT devices. Therefore, it is crucial to identify a balance between the client sampling ratio and resource consumption.

**Need for Resilience on Noisy Labels:** As demonstrated in Table 6, certain datasets exhibit high sensitivity to label errors, significantly deterring FL performance. Notably, both WISDM-W and Widar experience a drastic decrease in accuracy when faced with a 10% label noise ratio. Given the inevitability of noise in real FL deployments where private data is unmonitored except by the respective data owners, the development of label noise resilient techniques becomes crucial for achieving reliable FL performance. However, handling noisy labels is a much less explored topic in FL. Pioneering work on this topic is still exploring the effects of noisy labels on the performance of FL (Feng et al., 2023; Zhang et al., 2023a). To mitigate the effect of noisy labels, for low label noise rates, techniques such as data augmentation and regularization have been shown to be effective in centralized training settings (Shorten & Khoshgoftaar, 2019; Zhang et al., 2017). If the label error rate is high, techniques such as knowledge distillation (Hinton et al., 2015), mixup Zhang et al. (2017), and bootstrapping (Reed et al., 2014) have been proposed. We expect such techniques or their variants would help in the FL setting.

**Need for Quantized Training:** Table 8 highlights the need for quantized training given the limited RAM resources on representative IoT devices. From our results, we observe that FP16 quantization does not affect accuracy drastically unless normalization layers are present like batch normalization (Jacob et al., 2018). To mitigate this issue, techniques such as incremental quantization (Zhou et al., 2017) or ternary weight networks (Li et al., 2016) have shown great performance in the centralized training setting. We expect such techniques or their variants would help in the FL setting as well.

## 5 CONCLUSION

In this paper, we presented FedAIoT, an FL benchmark for AIoT. FedAIoT includes eight datasets collected from a wide range of authentic IoT devices as well as a unified end-to-end FL framework for AIoT that covers the complete FL-for-AIoT pipeline. We have benchmarked the performance of the datasets and provided insights on the opportunities and challenges of FL for AIoT. Moving forward, we aim to foster community collaboration by launching an open-source repository for this benchmark, continually enriching it with more datasets, algorithms, and thorough analytical evaluations, ensuring it remains a dynamic and invaluable resource for FL for AIoT research.

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
