# A  APPENDIX

## A.1  DEFINITIONS

### A.1.1  DIRICHLET ALLOCATION FOR SIMULATING DATA HETEROGENEITY

We used Dirichlet distribution, which is often used to simulate data heterogeneity across different clients. We define Dirichlet distribution in the following way: Given a K-dimensional vector $\boldsymbol{\alpha} = (\alpha_1, \alpha_2, ..., \alpha_K)$ where each $\alpha_i > 0$, a random vector $\boldsymbol{X} = (X_1, X_2, ..., X_K)$ follows a Dirichlet distribution, denoted as $\boldsymbol{X} \sim \text{Dir}(\boldsymbol{\alpha})$, if its probability density function (PDF) is given by:

$$f(\boldsymbol{x}; \boldsymbol{\alpha}) = \frac{1}{\text{B}(\boldsymbol{\alpha})} \prod_{i=1}^{K} x_i^{\alpha_i - 1}$$

subject to the conditions $x_i \geq 0$ for all $i$ and $\sum_{i=1}^{K} x_i = 1$. Here, $\text{B}(\boldsymbol{\alpha})$ is the multinomial beta function, defined as:

$$\text{B}(\boldsymbol{\alpha}) = \frac{\prod_{i=1}^{K} \Gamma(\alpha_i)}{\Gamma\left(\sum_{i=1}^{K} \alpha_i\right)}$$

where $\Gamma(\cdot)$ is the gamma function.

## A.2  SUPPLEMENTARY DATASET DETAILS

### A.2.1  WISDM

The WISDM dataset comprises raw accelerometer and gyroscope data collected from 51 subjects performing 18 activities for three minutes each. Data were gathered at a 20Hz sampling rate from both a smartphone (Google Nexus 5/5x or Samsung Galaxy S5) and a smartwatch (LG G Watch). Data for each device and sensor type are stored in different directories, resulting in four directories overall. Each directory contains 51 files, each corresponding to a subject. The data entry format is: <subject-id, activity-code, timestamp, x, y, z>. Separate files for the gyroscope and accelerometer readings are provided and are later combined by matching timestamps. Subject ID is given from 1600 to 1650 and the activity code is an alphabetical character between 'A' and 'S' excluding 'N'. The timestamp is in Unix time. The code to read and partition the data into 10s segments is provided by our benchmark. The input shape of the processed data is $200 \times 6$. The original dataset is available at `https://archive.ics.uci.edu/dataset/507/wisdm+smartphone+and+smartwatch+activity+and+biometrics+dataset`

### A.2.2  UT-HAR

The UT-HAR dataset was collected using the Linux 802.11n Channel State Information (CSI) Tool for the task of Human Activity Recognition (HAR). The original data consist of two file types: "input" and "annotation". "input" files contain Wi-Fi CSI data. The first column indicates the timestamp in Unix. Columns 2-91 represent amplitude data for 30 subcarriers across three antennas, and columns 92-181 contain the corresponding phase information. "annotation" files provide the corresponding activity labels, serving as the ground truth for HAR. In our benchmark, only amplitude is used. The final samples are created by taking a sliding window of size 250 where each sample consists of amplitude information across three antennas and from 30 subcarriers and has shape $3 \times 30 \times 250$. The original dataset is available at `https://github.com/ermongroup/Wifi_Activity_Recognition/tree/master`

### A.2.3  WIDAR

The Widar dataset (Widar3.0) was collected with a system comprising one transmitter and three receivers, all equipped with Intel 5300 wireless NICs. The system uses the Linux CSI Tool to record the Wi-Fi data. Devices operate in monitor mode on channel 165 at 5.825 GHz. The transmitter

broadcasts $1,000$ Wi-Fi packets per second while receivers capture data using their three linearly arranged antennas. In our benchmark, we use the processed body velocity profile (BVP) features extracted from the dataset. The size of each data sample after processing is $22 \times 20 \times 20$ consisting of 22 samples over time each having 20 BVP features each in both $x$ and $y$ directions. The raw dataset is available for download at `http://tns.thss.tsinghua.edu.cn/widar3.0/index.html`

### A.2.4 VISDRONE

The VisDrone dataset was collected by the AISKYEYE team at Tianjin University, China. It comprises 288 video clips with 261,908 frames and 10,209 static images captured by cameras mounted on drones at 14 different cities in China in diverse environments, scenarios, weather, and lighting conditions. The frames were manually annotated with over 2.6 million bounding boxes of common targets like pedestrians, cars, and bicycles. Additional attributes like scene visibility, object class, and occlusion are also provided for enhanced data utilization. The dataset is available at `https://github.com/VisDrone/VisDrone-Dataset`

### A.2.5 CASAS

The CASAS dataset is a collection of data generated in smart home environments, where intelligent software uses sensors deployed at homes to monitor resident activities and conditions within the space. The CASAS project considers environments as intelligent agents and employs custom IoT hardware known as Smart Home in a Box (SHiB), which encompasses the necessary sensors, devices, and software. The sensors in SHiB perceive the status of residents and their surroundings, and through controllers, the system acts to enhance living conditions by optimizing comfort, safety, and productivity. The CASAS dataset includes the date (in yyyy-mm-dd format), time (in hh:mm:ss.ms format), sensor name, sensor readings, and an activity label in string format. The data were collected in real-time as residents go about their daily activities. The code to extract categorical sensor readings to create input sequences and labels is provided in our benchmark. The CASAS dataset can be downloaded from https://casas.wsu.edu/datasets/.

### A.2.6 AEP

The AEP dataset, collected over 4.5 months, comprises readings taken every 10 minutes from a ZigBee wireless sensor network monitoring house temperature and humidity. Each wireless node transmitted data around every 3.3 minutes, which were then averaged over 10-minute periods. Additionally, energy data was logged every 10 minutes via m-bus energy meters. The dataset includes attributes such as date and time (in year-month-day hour:minute:second format), the energy usage of appliances and lights (in Wh), temperature and humidity in various rooms including the kitchen ($T_1$, $RH_1$), living room ($T_2$, $RH_2$), laundry room ($T_3$, $RH_3$), office room ($T4$, $RH_4$), bathroom ($T_5$, $RH_5$), ironing room ($T_7$, $RH_7$), teenager room ($T_8$, $RH_8$), and parents room ($T_9$, $RH_9$), and temperature and humidity outside the building ($T_6$, $RH_6$) - all with temperatures in Celsius and humidity in percentages. Additionally, weather data from Chievres Airport, Belgium was incorporated, consisting of outside temperature (To in Celsius), pressure (in mm Hg), humidity ($RH_{out}$ in %), wind speed (in m/s), visibility (in km), and dew point ($T_{dewpoint}$ in °C). The dataset is available at https://archive.ics.uci.edu/dataset/374/appliances+energy+prediction.

### A.2.7 EPIC-SOUNDS

As an extension of the EPIC-KITCHENS-100 dataset, the EPIC-SOUNDS dataset focuses on annotating distinct audio events in the videos of EPIC-KITCHENS-100. The annotations include the time intervals during which each audio event occurs, along with a text description explaining the nature of the sound. Given the variation in video lengths in the dataset, which range from 30 seconds to 1.5 hours, the videos are segmented into clips of 3-4 minutes each to make the annotation process more manageable. In order to ensure that annotators concentrate solely on the audio aspects, only the audio stream is provided to them. This decision is taken to prevent bias that could be introduced by the visual and contextual elements in the videos. Additionally, annotators are given access to the plotted audio waveforms. These visual representations of the audio data help the annotators by guiding them in pinpointing specific sound patterns, thus making the annotation process more efficient and targeted.

The EPIC-SOUNDS dataset can be extracted from the EPIC-KITHENS-100 dataset with the GitHub repo at https://github.com/epic-kitchens/epic-sounds-annotations. The extracted audio data in the form of HDF5 file format can also be requested from uob-epic-kitchens@bristol.ac.uk.

## A.3 HYPERPARAMETERS

**Hyperparameters for Table 4.** For WISDM-W, the learning rate for centralized training was 0.01 and we trained for 200 epochs with batch size 64. For FedAvg, in both low and high data heterogeneity scenarios, we used a client learning rate of 0.01 and trained for 400 communication rounds with batch size 32. For FedOPT, in both low and high data heterogeneity scenarios, we used a client learning rate of 0.01 and a server learning rate of 0.01. We also trained for 400 communication rounds. For WISDM-P, the learning rate for centralized training was 0.01 and we trained for 200 epochs with batch size 128. For FedAvg, in both low and high data heterogeneity scenarios, we used a client learning rate of 0.008 and trained for 400 communication rounds with batch size 32. For FedOPT, in both low and high data heterogeneity scenarios, we used a client learning rate of 0.01 and a server learning rate of 0.01. We also trained for 400 communication rounds. For UT-HAR and Widar, the learning rate for centralized training was 0.001 and the number of epochs was 500 and 200 for UT-HAR and Widar respectively with a batch size of 32. For both low and high data heterogeneity in both FedAvg and FedOPT, the client learning rate was 0.01 and the server learning rate for FedAvg and FedOPT was 1 and 0.01 respectively. The number of communication rounds was 1200 and 900 for UT-HAR and Widar respectively with a batch size of 32. For VisDrone, we used a cosine learning rate scheduler with $T_0 = 10, T_{mult} = 2$ and trained for 200 epochs with a learning rate of 0.1 and batch size 12. For all the experiments on VisDrone, the client learning rate was also 0.1 and the batch size was 12. For FedOPT, the server learning rate was 0.1. For CASAS, the centralized learning rate was 0.1 with batch size 128. For the federated setting, the client learning rate was 0.005, and the batch size was 32. We trained for 400 rounds. For FedOPT, the server learning rate was 0.01. For AEP, the learning rate for centralized training was 0.001 and the batch size was 32 and it was trained for 1200 epochs. For federated experiments, the client learning rate was 0.01, and the batch size was 32. For FedOPT, the server learning rate was 0.1. For EPIC-SOUNDS, for centralized training, the learning rate was 0.1 with batch size 512. The number of epochs was 120. For federated settings, we used a client learning rate of 0.1 and batch size 32. For FedOPT, the server learning rate was 0.01.

**Hyperparameters for Table 5.** The setup for all the datasets with 10% client sampling rate is the same as that of Table 4 under high data heterogeneity. For the 30% client sampling rate, the hyperparameters were kept the same as that of the 10% client sampling rate experiments, with the exception of CASAS, where the learning rate was set to 0.15.

**Hyperparameters for Table 6.** The hyperparameters were the same as that of Table 4 with 10% sampling rate under high data heterogeneity scenario.

**Hyperparameters for Table 7.** The hyperparameters were same as that of Table 4 with 10% client sampling rate under high data heterogeneity scenario.

## A.4 MODEL ARCHITECTURES

### A.4.1 WISDM

For WISDM, we use a custom LSTM model that consists of an LSTM layer followed by a feed-forward neural network. The LSTM layer has an input dimension of 6 and a hidden dimension of 6. After the LSTM layer, the output is flattened and passed through a dropout layer with a rate of 0.2 for regularization. It then goes through a fully connected linear layer with an input size of $1,200$ (6 hidden units * 200 timesteps) and an output size of 128, followed by a ReLU activation function. Another dropout layer with a rate of 0.2 is applied before the final fully connected linear layer with an input size of 128 and an output size of 12.

### A.4.2 UT-HAR

For UT-HAR, we use a ResNet-18 model with custom architecture designed for the Wi-Fi based Human Activity Recognition (HAR) task. The model consists of an initial convolutional layer that reshapes the input into a 3-channel tensor followed by the main ResNet architecture with 18 layers.

This main architecture includes a series of convolutional blocks with residual connections, Group Normalization layers, ReLU activations, and max-pooling. Finally, there is an adaptive average pooling layer followed by a fully connected layer that outputs the class probabilities. The model utilizes 64 output channels in the initial layer and doubles the number of channels as it goes deeper. The last fully connected layer has 7 output units corresponding to the number of classes for the UT-HAR task.

### A.4.3  WIDAR

For Widar, we also use a custom ResNet-18 model tailored for the Widar dataset. The model starts by reshaping the 22-channel input to 3 channels using two convolutional transpose layers, followed by a convolutional layer with 64 filters, Group Normalization, ReLU activation, and max-pooling. The core of the model consists of four layers of residual blocks (similar to the standard ResNet18) with 64, 128, 256, and 512 filters. Each basic block within these layers contains two convolutional layers, Group Normalization, and ReLU activations. Finally, an adaptive average pooling layer reduces spatial dimensions to $1 \times 1$, followed by a fully connected layer to output class scores.

### A.4.4  VISDRONE

For VisDrone, we use the default YOLOv8n model from Ultralytics library. YOLOv8n is the smallest YOLOv8 model variant with the three scale parameters: depth, width, and the maximum number of channels set to 0.33, 0.25, and 1024 respectively.

### A.4.5  CASAS

For CASAS, we use a BiLSTM neural network which is composed of an embedding layer, a bidirectional LSTM, and a fully connected layer. The embedding layer takes input sequences with dimensions equal to the input dimension and converts them to dense vectors of size 64. The bidirectional LSTM layer has an input size equal to 64, the same number of hidden units, and processes the embedded sequences in both forward and backward directions. The output of the LSTM layer is connected to a fully connected layer with an input size of 128 (to account for the bidirectional LSTM concatenation) and outputs the logits for 12 activities in the CASAS dataset.

### A.4.6  AEP

For AEP, we use a custom multi-layer perceptron (MLP) neural network with an architecture comprising five hidden layers and an output layer. The input layer accepts 18 features and passes them through a linear transformation to the first hidden layer with 210 units. Each of the following hidden layers progressively scales the number of units by factors of 2 and 4 and then scales down. Specifically, the sizes of the hidden layers are 210, 420, 840, 420, and 210 units respectively. Each hidden layer uses a ReLU activation function followed by a dropout layer with a dropout rate of 0.3 for regularization. The output layer has a single unit, and the output of the network is obtained by passing the activations of the last hidden layer through a final linear transformation.

### A.4.7  EPIC-SOUNDS

For EPIC-SOUNDS, we again use a custom ResNet-18 model which consists of a stack of convolutional layers followed by batch normalization and ReLU activation. The architecture begins with a $7 \times 7$ convolutional layer with stride 2, followed by a max pooling layer. Then, it contains four blocks, each comprising a sequence of basic blocks with a residual connection; specifically, each block contains two basic blocks, with output channel sizes of 64, 128, 256, and 512 respectively. Each basic block comprises two sets of 3x3 convolutional layers, each followed by batch normalization and ReLU activation. The first convolutional layer in the basic block has a stride of 2 in the second, third, and fourth blocks. Finally, the model has an adaptive average pooling layer, which reduces the spatial dimensions to 1x1, followed by a fully connected layer with an output size of 44 classes.

Table 8: Representative IoT devices for each dataset.

| Application | Dataset | IoT Platform | Representative Devices | Hardware RAM Size |
|---|---|---|---|---|
| | WISDM-W | Smartwatch | Apple Watch 8 | 512 MB to 1 GB |
| Activity Recognition | WISDM-P | Smartphone | iPhone 14 | 6 GB |
| | UT-HAR | Wi-Fi Router | TP-Link AX1800 | 64 MB to 1 GB |
| Gesture Recognition | Widar | Wi-Fi Router | TP-Link AX1800 | 64 MB to 1 GB |
| Independent Living | CASAS | Smart Home | Raspberry Pi 4 | 1 GB to 8 GB |
| Energy Prediction | AEP | Smart Home | Raspberry Pi 4 | 1 GB to 8 GB |
| Objective Detection | VisDrone | Drone | Dji Mavic 3 + Raspberry Pi 4 | 1 GB to 8 GB |
| Augmented Reality | EPIC-SOUNDS | Head-mounted Device | GoPro / AR Headset | 1 GB to 8 GB |

Table 9: Communication overheads under different quantization.

| | Communication cost per round | | | | | | | |
|---|---|---|---|---|---|---|---|---|
| | WISDM-W | WISDM-P | UT-HAR | Widar | VisDrone | CASAS | AEP | EPIC-SOUNDS |
| FP32 | 595.9KB | 595.9KB | 42.67MB | 42.69MB | 11.349MB | 750.9KB | 3.391KB | 42.95MB |
| FP16 | 297.95KB | 297.95KB | 21.335MB | 21.345MB | 5.6745MB | 375.45KB | 1.6955KB | 21.475MB |