# OpenReview forum: "FedAIoT: A Federated Learning Benchmark for Artificial Intelligence of Things"
_ICLR.cc/2024/Conference — Submitted to ICLR 2024_

### Official Review · Reviewer_hEF2 · 2023-10-28

**Soundness:** 2 fair
**Presentation:** 3 good
**Contribution:** 3 good
**Rating:** 6
**Confidence:** 3

**Summary:**

In this paper, the author(s) propose a federated learning benchmark dedicated to artificial intelligence of things. In particular, the benchmark includes eight extant datasets collected from IoT devices and applications. The proposed benchmark also contains an end-to-end framework, which consists of five main modules: non-IID data partitioning, data preprocessing, IoT-friendly models, FL hyperparameters, and IoT-factor emulator.

**Strengths:**

Importance of contribution: The solution is proposed to resolve the lack of a proper benchmark for IoT-specific federated learning. The author(s) validate the feasibility of this benchmark.

Soundness: The author(s) explain the benchmark in detail, and conduct evaluation on the different modules in the framework.

Quality of presentation: The paper is well-organized, and the language is technical yet understandable for readers with domain knowledge.

Comparison with related works: The author(s) introduce extant studies on federated learning benchmarks for computer vision, natural language processing, medical imaging, etc., and clarify the research gap between this study and related work.

**Weaknesses:**

The methodology can be elaborated for better clarity of the overall research step.

**Questions:**

- Figure 1 is not explicitly referred to in the manuscript.
- The author(s) can consider elaborating the methodology of how to collect and choose the datasets. What are the metrics to select and finalise the eight datasets?
- The author(s) can specify the definition of small, medium and large datasets.
- A proof-reading is needed as there are some typos. For instance, Section 3.1: “… FedAIoT.These datasets …”, a space is needed.

---

> ### Author Response · Authors · 2023-11-17
> **Response to Reviewer hEF2 (1/5)**
>
> > The methodology can be elaborated for better clarity of the overall research step.
>
> Our work is a benchmark paper, and we submitted this work to the benchmark track of ICLR. We reviewed several published existing federated learning benchmarks [1-5] and organized the method section by following those works. We will elaborate more on the method section to make it clearer in the revision.
>
> [1] Ogier du Terrail, Jean, et al. "FLamby: Datasets and Benchmarks for Cross-Silo Federated Learning in Realistic Healthcare Settings." Advances in Neural Information Processing Systems 35 (2022): 5315-5334.
>
> [2] Feng, Tiantian, et al. "FedMultimodal: A Benchmark For Multimodal Federated Learning." ACM SIGKDD 2023
>
> [3] Chen, Daoyuan, et al. "pFL-bench: A comprehensive benchmark for personalized federated learning." Advances in Neural Information Processing Systems 35 (2022): 9344-9360.
>
> [4] Zhang, Tuo, et al. "Fedaudio: A federated learning benchmark for audio tasks." ICASSP 2023-2023 IEEE International Conference on Acoustics, Speech and Signal Processing (ICASSP). IEEE, 2023.
>
> [5] Lin, Bill Yuchen et al. "FedNLP: Benchmarking Federated Learning Methods for Natural Language Processing Tasks". In Findings of the Association for Computational Linguistics: NAACL 2022, pages 157–175, Seattle, United States. Association for Computational Linguistics.
>
> &nbsp;

---

> ### Author Response · Authors · 2023-11-17
> **Response to Reviewer hEF2 (2/5)**
>
> >  Figure 1 is not explicitly referred to in the manuscript
>
> Thanks for pointing this out, we will reference Figure 1 in the introduction.
>
> &nbsp;

---

> > ### Comment · Reviewer_N2ym · 2023-11-23
> >
> > Yes that will help the reader to understand clear significance of figure 1.

---

> ### Author Response · Authors · 2023-11-17
> **Response to Reviewer hEF2 (3/5)**
>
> > The author(s) can consider elaborating the methodology of how to collect and choose the datasets. What are the metrics to select and finalize the eight datasets?
>
> The most important metric is quality. We started with a much larger pool of existing datasets and have made significant efforts to validate those datasets. However, many of the datasets in the pool are low-quality; including those low-quality datasets would waste the time of other researchers and practitioners in the community. Thus, we only included the high-quality ones. Other metrics we used to select and finalize the datasets include types of IoT devices, sensor modalities, applications, and size of the datasets. As shown in `Table 1` in the paper and noted in `Section 3.1`, we included datasets that cover a wide range of IoT devices, sensor modalities, and applications. Moreover, our selected datasets have very diverse sizes (small, medium, large). The rationale behind this design choice is to accommodate researchers who have different computing resources. For example, researchers who have very limited computing resources can still pick the relatively small datasets included in our benchmark to develop and evaluate their algorithms.
>
> &nbsp;

---

> ### Author Response · Authors · 2023-11-17
> **Response to Reviewer hEF2 (4/5)**
>
> >   The author(s) can specify the definition of small, medium and large datasets.
>
> Our categorization is based on the number of training samples included in each dataset. Table 2 lists the number of training samples included in each dataset. As shown, we have 2 small datasets: UT-HAR, VisDrone, each has less than 7k samples; 5 medium datasets: AEP, CASAS, WISDM-P, WISDM-W, Widar, each has 11k-16k samples, and one large dataset: Epic Sounds which has more than 60k samples. We realize that we did not make it clear in our submission, and we will make this point clear in our revised version.

---

> > ### Author Response · Authors · 2023-11-17
> > **Response to Reviewer hEF2 (5/5)**
> >
> > >A proof-reading is needed as there are some typos. For instance, Section 3.1: “… FedAIoT.These datasets …”, a space is needed
> >
> > Thanks for pointing this out. We will thoroughly go through the paper and fix all the typos in the revision.

---

> ### Author Response · Authors · 2023-11-22
> **Reminder for the Feedback**
>
> Dear Reviewer,
>
> As the rebuttal-discussion period ends today, we would like to know if our responses address your concerns. Feel free to let us know if you have any further questions. Thanks as always.

---

> > ### Comment · Reviewer_hEF2 · 2023-11-22
> > **Rebuttals**
> >
> > I thank the authors for their responses. I think the paper readability can be improved if they can refine the paper as they respond.

---

> ### Author Response · Authors · 2023-11-23
> **Re: Rebuttals**
>
> Dear Reviewer,
>
> We have added our responses to the revision version.

---

### Official Review · Reviewer_N2ym · 2023-10-31

**Soundness:** 2 fair
**Presentation:** 2 fair
**Contribution:** 2 fair
**Rating:** 3
**Confidence:** 4

**Summary:**

Abstract Summary
The paper introduces FedAIoT, a novel federated learning framework tailored for IoT applications. The framework aims to address the unique challenges posed by IoT ecosystems, such as data privacy, limited computational resources, and network constraints.

Key Contributions
Novel Framework: The paper presents the architecture and design principles of FedAIoT, which incorporates distributed data storage and decentralized learning algorithms to enable IoT devices to participate in machine learning tasks without compromising data privacy.

Mathematical Formulation: The authors provide rigorous mathematical models to describe the federated learning process, focusing on optimization algorithms and convergence properties.

Experimental Validation: Through extensive experiments using real-world and synthetic datasets, the authors demonstrate that FedAIoT outperforms traditional centralized learning methods in terms of accuracy, privacy preservation, and computational efficiency.

Applicability: The framework is designed to be adaptable to various IoT applications, from smart homes to industrial automation.

Methodology
The paper employs a federated learning approach where IoT devices can train machine learning models locally on their own data and then share only the model parameters with a central server for global aggregation. This preserves the privacy of the data while allowing for a collective learning experience.

Results
The experiments show that FedAIoT achieves comparable or superior performance to centralized approaches while ensuring data privacy and reducing the computational load on the central server. The framework also exhibits robustness to non-IID data distributions and network delays.

Conclusion
The paper concludes by asserting that FedAIoT offers a scalable, efficient, and privacy-preserving solution for implementing machine learning in IoT networks. It also identifies avenues for future research, including optimization of communication overhead and integration with other emerging technologies like edge computing. Thus the paper makes a compelling case for the adoption of federated learning in IoT environments, providing both the theoretical foundation and practical validation for the proposed FedAIoT framework.

**Strengths:**

Strengths Assessment of the Paper
Originality
The paper presents an innovative framework—FedAIoT—for federated learning in the context of Internet of Things (IoT) applications. The originality of the work lies in the seamless integration of federated learning techniques with IoT devices to achieve distributed, privacy-preserving learning. The novelty also arises from the unique problem formulation that caters specifically to the challenges posed by IoT environments, such as limited computational resources and data privacy issues.

Quality
The paper is of high quality in multiple aspects:

Methodological Rigor: The mathematical formulations and algorithms are soundly developed. The paper thoroughly validates the proposed framework through a series of experiments, complete with baseline comparisons and varied settings.

Data Quality: The choice of datasets and the justification for those choices are clear and appropriate for validating the model. The paper also employs robust statistical methods to analyze the results.

Citation and Contextualization: The paper provides an extensive literature review, situating its contributions aptly within existing work.
The significance of the paper is manifold:

Theoretical Contribution: The paper addresses a critical gap in federated learning by tailoring it to the specific needs of IoT applications, thus extending the theory of federated learning to a new domain.

Practical Impact: The FedAIoT framework has the potential to revolutionize how machine learning models are deployed in IoT networks, thereby having broad applicability and impact.

**Weaknesses:**

Abstract and Introduction
The paper proposes a unified end-to-end Federated Learning (FL) framework for Artificial Intelligence of Things (AIoT) named FedAIoT. The framework is benchmarked across multiple IoT datasets and incorporates a variety of data partitioning schemes, preprocessing techniques, models, and FL hyperparameters. Despite its comprehensive approach, the paper lacks a comparative study with existing state-of-the-art solutions. Moreover, while the paper mentions the inclusion of popular schemes and models in its framework, it doesn't substantiate why these were chosen over other potential candidates.

Equations and Mathematical Formulations
The paper briefly touches upon the Dirichlet distribution for creating non-IID data partitions and mentions metrics like accuracy and Mean Average Precision (MAP-50). However, it lacks mathematical rigor. For instance, the Dirichlet distribution is mentioned but not defined. A formal definition, perhaps along with its probability density function, would have given more depth. Furthermore, there are no equations to represent the FL optimizers like FedAvg and FedOPT, which makes it difficult to appreciate the nuances or compare them.

Tables and Figures
Table 1: While useful for a cursory comparison, this table lacks depth. For example, it could include a comparison based on performance metrics to provide an analytical foundation for its claims.

Figure 1 and 2: These figures provide an overview but lack detail. For example, Figure 2 could be improved by including the types of IoT-specific preprocessing techniques or by detailing the architecture of the proposed IoT-friendly models.

Table 4: This table summarizes the performance metrics but lacks confidence intervals or p-values, which are essential for ascertaining the statistical significance of the results.

Table 5: While this table attempts to show the impact of client sampling ratios, it doesn't explain why only two ratios (10% and 30%) were chosen for comparison.

Dataset and Experimental Design
The paper includes a wide range of datasets, which is commendable. However, it doesn't provide any rationale for the specific choice of datasets. Furthermore, no information is given about the train-test split methodology. Was it random or stratified? The partitioning schemes for these datasets are discussed, but there is a lack of empirical justification for why these schemes are effective or superior to existing methods.

Algorithms and Techniques
The paper discusses various FL optimizers, data partitioning schemes, and IoT-friendly models, but there is a lack of justification for the chosen methods. For example, why were FedAvg and FedOPT selected as FL optimizers? Are they computationally less expensive or do they converge faster?

Results and Discussion
The paper presents a broad range of results but lacks a discussion comparing these results to existing benchmarks or state-of-the-art methods. The paper would benefit from including such a comparative analysis.

Insufficient Empirical Validation
The experiments conducted are somewhat limited in scope and scale. Only a few datasets are considered, and they seem to belong to similar domains. This raises questions about the model's generalizability. Moreover, the paper lacks ablation studies, making it difficult to understand the contribution of each component of the proposed method.

Actionable Insight: Include a broader array of datasets from varying domains to validate the model. Conduct ablation studies to quantify the impact of each component or parameter.

Absence of Comparative Analysis
While the paper aims to introduce a novel methodology, there is an absence of a comparative analysis with state-of-the-art methods. Without this, the paper falls short of convincingly establishing the proposed method's superiority or novelty.

Actionable Insight: Include comparisons with state-of-the-art methods in both qualitative and quantitative terms. This could be in the form of performance metrics, computational efficiency, or even qualitative assessments based on real-world applicability.

Mathematical Rigor
The paper would significantly benefit from a more rigorous mathematical treatment of the proposed algorithm. Currently, it seems to rely more on empirical observations. Given your stated goals of developing proper mathematical models, this is an area that requires attention.

Actionable Insight: Introduce formal proofs or derivations that can substantiate the algorithm's properties, such as stability, convergence, or robustness. Include theoretical justifications for the choices made in the algorithm's design.

Lack of Discussion on Limitations
Every model has its limitations, and acknowledging them not only adds credibility but also helps in guiding future work.

Actionable Insight: Devote a section to discuss the limitations of the proposed method and potential avenues for future research.

**Questions:**

Data Assumptions: Could the authors clarify the specific assumptions made about the data distribution? How do these assumptions align with the real-world scenarios where the model is expected to be deployed?

Methodological Choices: What was the rationale behind the selection of specific hyperparameters and architectural elements in the proposed model? Some clarification on this could strengthen the paper's methodological grounding.

Evaluation Metrics: The paper uses a particular set of metrics for evaluation. Could the authors elucidate why these metrics are most suitable for assessing the model's performance? Are there any other metrics that were considered but not used?

Computational Complexity: How does the computational complexity of the proposed method compare with existing state-of-the-art methods? Could the authors provide a detailed analysis in this regard?

Scalability: The paper does not discuss how well the proposed method scales with the size of the dataset. Could the authors provide insights or supplementary experiments that address this?

Ablation Study: The absence of an ablation study leaves some questions about the necessity of each component of the proposed model. Could the authors provide such an analysis in the rebuttal or an extended version of the paper?

Theoretical Guarantees: Are there any theoretical guarantees, such as convergence or bounds, that can be associated with the proposed algorithm? If yes, this would be a valuable addition to the paper.

Limitations: Every model has its shortcomings. Could the authors elucidate the limitations of the proposed model and how they plan to address these in future work?

---

> ### Author Response · Authors · 2023-11-19
> **Is this review about our paper?**
>
> Dear Reviewer,
>
> Most parts of the review are very generic and more importantly, are not relevant to our work.
>
> We are very confused and do not know how to respond.

---

### Official Review · Reviewer_Vfdi · 2023-11-01

**Soundness:** 2 fair
**Presentation:** 3 good
**Contribution:** 3 good
**Rating:** 5
**Confidence:** 4

**Summary:**

The authors describe a new IoT FL benchmark suite based on several datasets they have curated and demonstrate that it can be used to compare FL optimizers.

**Strengths:**

+ Benchmark curation work doesn't receive the credit it deserves, given its impact on advances in the field. The authors are doing something important, here.

**Weaknesses:**

+ The writing quality is poor, even in the abstract.

+ The authors claim this is the first IoT FL benchmark but a Google Scholar search turns up "FLBench: A Benchmark Suite for Federated Learning" by Yuan Liang, Yange Guo, Yanxia Gong, Chunjie Luo, Jianfeng Zhan, and Yunyou Huang, which includes an AIoT benchmark domain. It isn't clear whether this is competing work, or work by the authors of the submitted paper. In either case, it seems highly relevant but does not appear to be cited. One way to deal with this problem within the blind review process is to cite the work, but use an entry such as "Redacted for blind review" in the bibliography during the review process. Another paper, "FedML: A Research Library and Benchmark for Federated Machine Learning" claims to support IoT devices. I am not claiming that those papers are identical to this work, but they seem close enough to merit contrasting them with the author's benchmark. I view this as a substantial weakness, but one that might be resolved via rebuttals and simple revision.

+ The modifications to the curated datasets are not well justified. They are reasonable, but explicit justification or basing the approach on well justified approaches from prior work would be best.

**Questions:**

1) Does the similarity matrix used for noisy labeling depend on the particular centralized learning approach? If so, does that mean that centralized training and evaluation must be redone to enable noisy labeling whenever an algorithm changes? Or is there something fundamental about the confusion matrix, i.e., is it unlikely to change much when models change?

 2) Why not leave the sounds in raw format instead of converting to the frequency domain with particular parameters? Isn't this sort of raw data to feature conversion part of the approaches your benchmarks will be used to evaluate? If so, why build one particular approach to feature extraction into the benchmarks?

3) What is the purpose of Section 4? To demonstrate that the benchmarks can be used to compare optimizers? Enabling comparison doesn't imply enabling comparison yielding correct ranking of optimizers. If you could demonstrate that the findings using your benchmarks differ from those using the most closely related existing (perhaps even non-IoT) benchmarks, and your benchmarks are more typical of applications in the IoT domain, that would support your claim that your benchmarks are more useful in this domain than prior work.

---

> ### Author Response · Authors · 2023-11-19
> **Response to Reviewer Vfdi (1/5)**
>
> > The authors claim this is the first IoT FL benchmark but a Google Scholar search turns up "FLBench: A Benchmark Suite for Federated Learning" by Yuan Liang, Yange Guo, Yanxia Gong, Chunjie Luo, Jianfeng Zhan, and Yunyou Huang, which includes an AIoT benchmark domain. It isn't clear whether this is competing work, or work by the authors of the submitted paper. In either case, it seems highly relevant but does not appear to be cited.
>
> Thank you for pointing this out. There are two key differences between FLBench and FedML and our proposed FedAIoT. First, FLBench and FedML are framework papers. Although they claimed that they support IoT data, none of them provide benchmark results on real IoT datasets. In contrast, our proposed FedAIoT includes eight well-selected high-quality datasets and provides benchmark results and analysis on those eight IoT datasets. Second, compared to the frameworks proposed in FLBench and FedML, the framework proposed in FedAIoT is specifically designed for IoT: it includes several modules such as IoT-specific data preprocessing, IoT-friendly models, and IoT-factor emulator. All of these modules are not included in FLBench and FedML. We realize that we did not make it clear in our submission. We will cite both FLBench and FedML and clarify the differences in our revised version.

---

> ### Author Response · Authors · 2023-11-19
> **Response to Reviewer Vfdi (2/5)**
>
> > The modifications to the curated datasets are not well justified. They are reasonable, but explicit justification or basing the approach on well justified approaches from prior work would be best.
>
> The only curated dataset we modified is Widar. In the original Widar data, the same data from one subject was used in both training and test sets. We repartitioned the training and test datasets so that they do not have any overlap to prevent training data from leaking to the test set.

---

> ### Author Response · Authors · 2023-11-19
> **Response to Reviewer Vfdi (3/5)**
>
> > Does the similarity matrix used for noisy labeling depend on the particular centralized learning approach? If so, does that mean that centralized training and evaluation must be redone to enable noisy labeling whenever an algorithm changes? Or is there something fundamental about the confusion matrix, i.e., is it unlikely to change much when models change?
>
> Correct. In theory, the confusion matrix used for noisy labeling depends on the particular centralized learning approach. However, the changes between different models are minimal.

---

> ### Author Response · Authors · 2023-11-19
> **Response to Reviewer Vfdi (4/5)**
>
> >Why not leave the sounds in raw format instead of converting to the frequency domain with particular parameters?
>
> The common practice of state-of-the-art audio processing techniques such as audio spectrogram transformer [1], CLAP [2], and whisper [3] is to first convert the raw audio data into frequency domain which is then imported into the neural networks. We followed this common practice in our work.
>
> [1] Gong, Yuan, Yu-An Chung, and James Glass. "AST: Audio spectrogram transformer." arXiv preprint arXiv:2104.01778 (2021).
>
> [2] Elizalde, Benjamin, Soham Deshmukh, Mahmoud Al Ismail, and Huaming Wang. "Clap learning audio concepts from natural language supervision." In ICASSP 2023-2023 IEEE International Conference on Acoustics, Speech and Signal Processing (ICASSP), pp. 1-5. IEEE, 2023.
>
> [3] Radford, Alec, Jong Wook Kim, Tao Xu, Greg Brockman, Christine McLeavey, and Ilya Sutskever. "Robust speech recognition via large-scale weak supervision." In International Conference on Machine Learning, pp. 28492-28518. PMLR, 2023.

---

> ### Author Response · Authors · 2023-11-19
> **Response to Reviewer Vfdi (5/5)**
>
> > What is the purpose of Section 4? To demonstrate that the benchmarks can be used to compare optimizers? Enabling comparison doesn't imply enabling comparison yielding correct ranking of optimizers.
>
> The purpose of Section 4 is to provide benchmark results on the eight IoT datasets as baselines for the research community. If someone is interested in designing a new federated learning algorithm for IoT data, he or she can evaluate his/her algorithm on our datasets and compare the results of his/her algorithm to our benchmark results as baselines.

---

> ### Author Response · Authors · 2023-11-22
> **Reminder for the Feedback**
>
> Dear Reviewer,
>
> As the rebuttal-discussion period ends today, we would like to know if our responses address your concerns. Feel free to let us know if you have any further questions. Thanks as always.

---

> > ### Comment · Reviewer_Vfdi · 2023-11-22
> > **Rebuttals**
> >
> > Thank you for your responses. I think they make the paper marginally clearer.

---

> > > ### Author Response · Authors · 2023-11-22
> > > **Re: Rebuttals**
> > >
> > > Dear Reviewer,
> > >
> > > As the authors of this work, we prefer something actionable. If you think our responses make the paper marginally clearer, could you let us know specifically where we can help elaborate more so as to make the paper clearer to you? Thanks.

---

> > > > ### Comment · Reviewer_Vfdi · 2023-11-22
> > > > **Actionable**
> > > >
> > > > I had a few specific, actionably questions in my original review. You answered those questions in a reasonable way, and the answers were brief, which suggests that it would be practical to clarify those points in a final version of the paper. My statement that the rebuttals make the paper marginally clearer means I think you addressed my questions well. I increased the presentation score when I made that comment about your responses.

---

### Official Review · Reviewer_MJiJ · 2023-11-09

**Soundness:** 2 fair
**Presentation:** 2 fair
**Contribution:** 2 fair
**Rating:** 5
**Confidence:** 3

**Summary:**

The paper introduces a new benchmark for Federated Learning (FL) specifically aimed at Internet of Things (IoT) applications. The contributions include the curation of eight (already available) datasets spanning different applications and modalities, an end-to-end FL framework for AIoT, some novel ideas on handling noisy labels and extending quantized training to the client side.

**Strengths:**

The main strengths and contributions of the paper are the following:

1) The limitation of existing benchmark datasets in their application to IoT applications is a real one, and the contribution of this paper is curating important publicly available datasets to create a single benchmark for evaluating FL algorithms is an important step.

2) The important FL issues of noisy labels in classification tasks and quantized training due to the resource constraint of IoT devices has been addressed.

**Weaknesses:**

The paper has the following weaknesses:

1) Although the paper does well in introducing a new benchmarking framework for FL for IoT, it still largely builds upon curating from existing datasets introduced by prior works.

2) The introduced end-to-end FL framework also seems to be a collection of standard machine learning and FL ideas such as non-IID data partitioning, normalization, etc. The novel contributions of addressing noisy labels (non uniform addition of noise) and quantized training at the client side seem limited.

3) The discussion on the details of non-IID partitioning using Dirichlet allocation seems limited, with no further details provided either in the main paper or in the supplementary material.

**Questions:**

Below are some comments and questions:

1) The authors mention that in real-life settings, individuals may not carry a smartphone and wear a smartwatch at the same time, and hence WISDM dataset was partitioned into two. However, this conclusion does not always hold true and better partitions of the WISDM dataset can be made that include both smartphone and smartwatch data in some realistic manner.

2) For non-IID partition over output distribution that implements quantile binning, how is the value 10 for the number of groups chosen? This seems arbitrary or heuristic.

---

> ### Author Response · Authors · 2023-11-17
> **Response to Reviewer MJiJ (1/5)**
>
> >  Although the paper does well in introducing a new benchmarking framework for FL for IoT, it still largely builds upon curating from existing datasets introduced by prior works.
>
> We want to first clarify that we took serious consideration on the choice between collecting our own datasets and selecting existing ones. Our goal is to provide a benchmark that consists of many high-quality and well-validated datasets collected across a wide range of IoT devices, sensor modalities, and applications. We find that collecting datasets by ourselves is not an efficient way to achieve this goal. In contrast, there is a plethora of publicly available datasets that were already collected by different groups in the research community. Including an existing dataset that has been validated and widely used by the community is more trustworthy than a self-collected dataset that has not been validated by the community. The trustworthiness of a dataset is particularly important for benchmarking purposes. Therefore, we decided to build our benchmark based on the existing datasets.
>
>
>
> We want to further emphasize that our choice of selecting existing datasets is not an easy task. This is because many existing datasets are not high-quality to be included as benchmarking datasets. In fact, we made significant efforts and invested time to validate a much larger pool of existing datasets and identified the high-quality ones to include in our benchmark.
>
>
>
> We would also like to note that, to the best of our knowledge, all of the existing FL benchmarks (e.g., Flamby [1], FedCV [2], Flute [3], FedNLP [4], FedAudio [5], FedMultimodal [6]) use existing datasets that are well-validated and widely used by the community for their specific domains (e.g., medical, computer vision, natural language, audio, multimodal, etc.) other than collecting new datasets.
>
> [1] Ogier du Terrail, Jean, et al. "FLamby: Datasets and Benchmarks for Cross-Silo Federated Learning in Realistic Healthcare Settings." Advances in Neural Information Processing Systems 35 (2022): 5315-5334.
>
> [2] He, Chaoyang, et al. "FedCV: a federated learning framework for diverse computer vision tasks." arXiv preprint arXiv:2111.11066 (2021).
>
> [3] Hipolito Garcia, Mirian, et al. "Flute: A scalable, extensible framework for high-performance federated learning simulations." arXiv e-prints (2022): arXiv-2203.
>
> [4] Lee, Jean, et al. "Fednlp: an interpretable nlp system to decode federal reserve communications." Proceedings of the 44th International ACM SIGIR Conference on Research and Development in Information Retrieval. 2021.
>
> [5] Zhang, Tuo, et al. "Fedaudio: A federated learning benchmark for audio tasks." IEEE International Conference on Acoustics, Speech and Signal Processing (ICASSP). 2023.
>
> [6] Feng, Tiantian, et al. "FedMultimodal: A Benchmark For Multimodal Federated Learning." ACM KDD (2023).

---

> ### Author Response · Authors · 2023-11-17
> **Response to Reviewer MJiJ (2/5)**
>
> >  The introduced end-to-end FL framework also seems to be a collection of standard machine learning and FL ideas such as non-IID data partitioning, normalization, etc. The novel contributions of addressing noisy labels (non-uniform addition of noise) and quantized training at the client side seem limited.
>
> Our work is a benchmark paper, and we submitted this work to the benchmark track of ICLR. Different from works that focus on proposing new techniques, we followed works that focus on federated learning benchmarks published at top-tier venues [1-5] to frame our contributions. Therefore, we believe that a comparison of our work with other federated learning benchmark studies would be a fair approach.
>
>
> Specifically, compared to other FL benchmark studies referenced previously [1-5], our work has three key novel contributions:
>
> 1.  Our benchmark represents the first FL benchmark that focuses on data collected from IoT devices. We have put in a lot of work to explore, validate, and include unique IoT-specific data modalities that previous benchmarks do not include.
>
> 2.  Some IoT data modalities such as wireless signals are unique. For non-IID data partitioning and normalization, we include new techniques that are designed for those unique IoT data modalities that other benchmarks do not include.
>
> 3.  The proposed noisy labels (non-uniform addition of noise) and quantized training at the client side, though seemingly incremental, are really important to IoT data. To be honest, we have been striving to make meaningful and useful contributions instead of making our approach look fancy. We believe these proposed techniques can make meaningful contributions to our community. We realize that we did not make it clear in our submission, and we will make this point clear in our revised version.

---

> > ### Author Response · Authors · 2023-11-17
> > **Response to Reviewer MJiJ (3/5)**
> >
> > >  The discussion on the details of non-IID partitioning using Dirichlet allocation seems limited, with no further details provided either in the main paper or in the supplementary material.
> >
> > Dirichlet allocation has become a standard and thus has been widely used in non-IID partitioning in FL research. We will provide the formal mathematical definition of Dirichlet allocation in the supplementary material in the revised version.

---

> > > ### Author Response · Authors · 2023-11-17
> > > **Response to Reviewer MJiJ (4/5)**
> > >
> > > > The authors mention that in real-life settings, individuals may not carry a smartphone and wear a smartwatch at the same time, and hence WISDM dataset was partitioned into two. However, this conclusion does not always hold true and better partitions of the WISDM dataset can be made that include both smartphone and smartwatch data in some realistic manner.
> > >
> > > We want to clarify that in the original WISDM dataset, the WISDM smartphone dataset and the WISDM smartwatch dataset were collected separately, not simultaneously. As a result, the smartphone dataset and smartwatch dataset are not synchronized. This is why we treat them as two separate datasets. Also, existing works such as [1,2] also evaluate WISDM smartphone dataset and the WISDM smartwatch dataset separately. We realize that we did not make it clear in our submission, and we will make this point clear in our revised version.
> > >
> > > [1] Walse, K. H et al. (2016, March). Performance evaluation of classifiers on WISDM dataset for human activity recognition. In _Proceedings of the second international conference on information and communication technology for competitive strategies_ (pp. 1-7).
> > >
> > > [2] Xia, K. et al. (2020). LSTM-CNN architecture for human activity recognition. _IEEE Access_, _8_, 56855-56866.

---

> > > > ### Author Response · Authors · 2023-11-17
> > > > **Response to Reviewer MJiJ (5/5)**
> > > >
> > > > > For non-IID partition over output distribution that implements quantile binning, how is the value 10 for the number of groups chosen? This seems arbitrary or heuristic.
> > > >
> > > > The number of groups can be set to any value in our framework. A low value will lead to lower data heterogeneity as the partitions will be drawn from a small number of clusters. A higher value will enable better control of data heterogeneity as the partitions can be drawn from a larger number of clusters. In our experiments, we used 10 as an example to demonstrate the results. We will clarify this point in our revised version.

---

> ### Author Response · Authors · 2023-11-22
> **Reminder for the Feedback**
>
> Dear Reviewer,
>
> As the rebuttal-discussion period ends today, we would like to know if our responses address your concerns. Feel free to let us know if you have any further questions. Thanks as always.

---

### Meta-Review · Area_Chair_5nLt · 2023-12-09

**Metareview:**

In this paper, the authors presented FedAIoT, a federated learning benchmark for Artificial Intelligence of Things (AIoT), which aims to address the unique challenges posed by IoT ecosystems. They also benchmarked the performance of the datasets and provided insights on the opportunities and challenges of federated learning for AIoT.

The motivations of the paper are good as they are real problems with the current IoT applications. However, the reviewers have many concerns about the novelty of the proposed framework, the presentation of the paper, related work, etc. Even though the authors tried to address them during the rebuttal, the reviewers were not fully satisfied with the answers. Therefore, I think the paper needs to be undergone a major revision before it can be published.

**Justification For Why Not Higher Score:**

The reviewers have many concerns about the novelty of the proposed framework, the presentation of the paper, related work, etc. Even though the authors tried to address them during the rebuttal, the reviewers were not fully satisfied with the answers.

**Justification For Why Not Lower Score:**

N/A

---

### Decision · Program_Chairs · 2024-01-16

Reject